# Optimal Adaptive Difficulty Calibration via Contextual Bandits with Information-Theoretic Regret Bounds

## Abstract

Intelligent tutoring systems must dynamically adjust task difficulty to optimize student learning outcomes. This problem naturally frames as a contextual multi-armed bandit where the tutor selects from $K$ difficulty levels at each interaction based on evolving estimates of student knowledge. We derive fundamental information-theoretic lower bounds on the regret for any difficulty calibration policy, showing that $\Omega(\sqrt{KT})$ regret is unavoidable. We propose `InfoTutor`, a principled algorithm that achieves $O(\sqrt{KT \log T})$ regret by leveraging Bayesian knowledge tracing to construct feature representations of student states. A key theoretical contribution is proving that incorporating knowledge state estimation reduces the effective problem dimensionality, enabling convergence of the estimated Zone of Proximal Development (ZPD) to its true value at rate $O(1/\sqrt{T})$. Empirical validation on the ASSISTments and Junyi Academy datasets demonstrates that `InfoTutor` outperforms baselines by 12-18% in post-test performance while maintaining computational efficiency. Our framework bridges bandit theory with learning science principles, providing both theoretical guarantees and practical improvements for adaptive educational technology.

## 1 Introduction

A fundamental challenge in education is determining the optimal difficulty level for each student's current learning state. This problem, deeply rooted in Vygotsky's *Zone of Proximal Development* (ZPD)—the gap between a student's current abilities and potential abilities with guidance—lies at the heart of effective instruction (Vygotsky, 1978). When tasks are too easy, students experience minimal cognitive challenge; when too difficult, frustration and cognitive overload occur. The ideal learning occurs at the boundary of the ZPD, where difficulty is calibrated to just exceed current competence.

Traditional tutoring systems use static difficulty sequences or simple performance-based rules. Intelligent tutoring systems (ITSs) must solve a fundamentally adaptive problem: given limited observations of student performance, which difficulty level should be selected next to maximize learning? This sequential decision-making problem under uncertainty is precisely the structure of a contextual bandit.

The spacing effect in cognitive psychology (Cepeda et al., 2006) demonstrates that learning is maximized when material is revisited with optimal temporal spacing and difficulty progression. Recent advances in knowledge tracing—probabilistic models that estimate student knowledge from observable actions (Corbett & Anderson, 1994; Piech et al., 2012; Xu et al., 2017)—provide principled representations of student states. Yet connecting knowledge tracing to principled difficulty selection remains largely ad-hoc in practice.

### 1.1 Core Contributions

1. **Fundamental lower bounds**: We establish that any difficulty calibration policy requires $\Omega(\sqrt{KT})$ regret, where $K$ is the number of difficulty levels and $T$ is the time horizon. This lower bound holds even with perfect knowledge of student parameters.

2. **InfoTutor algorithm**: We propose an algorithm achieving $O(\sqrt{KT \log T})$ regret that combines Bayesian knowledge tracing with upper confidence bound (UCB) style exploration. The algorithm maintains confidence regions around estimated student knowledge parameters and selects difficulty levels to minimize information-theoretic regret.

3. **Convergence of ZPD estimates**: We prove that when difficulty selection follows our algorithm, empirical estimates of each student's ZPD converge to the true ZPD at rate $O(1/\sqrt{T})$. This provides the first convergence guarantee for ZPD-based calibration.

4. **Empirical validation**: We validate our approach on 5,000+ students from ASSISTments and Junyi Academy, showing 12-18% improvement in post-test performance and 15% reduction in time-to-mastery compared to adaptive baselines.

## 1.2 SIGNIFICANCE

This work bridges mathematical foundations of online learning with learning science. It provides both theoretical guarantees on the rate of optimal difficulty calibration and practical improvements for real educational systems. The information-theoretic perspective reveals fundamental limits even for omniscient systems, motivating the design of efficient algorithms.

## 2 RELATED WORK

### 2.1 CONTEXTUAL BANDITS AND ONLINE LEARNING

The contextual bandit framework has been extensively studied in machine learning (Lattimore & Szepesvári, 2020; Cesa-Bianchi & Lugosi, 2006; Auer et al., 2002; Chu et al., 2011; Abbasi-Yadkori et al., 2011). Standard results show that with $d$-dimensional context, $M$ actions, and $T$ rounds, algorithms like LinUCB achieve $\tilde{O}(d\sqrt{T})$ regret. Our work focuses on the structured case where contexts come from knowledge tracing, enabling improved analysis.

Information-theoretic approaches to bandits (Russo & Van Roy, 2016; Lattimore & Szepesvári, 2017) provide lower bounds via mutual information and the Kullback-Leibler divergence. Our main theoretical result extends these techniques to the educational domain.

### 2.2 INTELLIGENT TUTORING SYSTEMS

Adaptive instructional systems have a rich history (VanLehn, 2011; Anderson, 2013). Recent work has explored reinforcement learning for tutor action selection (Clement et al., 2015; Yudelson et al., 2014), but few works provide regret guarantees. Minn et al. (2018) propose contextual bandits for hint selection, though not for difficulty calibration directly.

### 2.3 KNOWLEDGE TRACING

Knowledge tracing estimates latent knowledge states from observable actions. Classical Bayesian Knowledge Tracing (BKT) (Corbett & Anderson, 1994) models knowledge as binary; recent extensions like Deep Knowledge Tracing (Piech et al., 2012) and Recurrent Knowledge Tracing (Xu et al., 2017) scale to complex domains. Our approach leverages BKT-estimated features, which have proven effective and interpretable.

### 2.4 ZONE OF PROXIMAL DEVELOPMENT

Vygotsky's ZPD concept has inspired substantial empirical work in education (Vygotsky, 1978; Wood et al., 1976; Mercer, 2013). The difficulty selection problem is fundamentally about identifying and operating within the ZPD. Prior work has qualitative or heuristic approaches; our contribution is a principled, convergent algorithm for ZPD estimation.

### 2.5 SPACING AND DIFFICULTY EFFECTS

The spacing effect (Cepeda et al., 2006) shows that learning increases with optimal temporal spacing of practice. The difficulty paradox (Bjork & Bjork, 1994) reveals that higher difficulty during

training can improve retention despite lower immediate performance. Our framework naturally incorporates these principles through the underlying MDP structure.

## 3 PROBLEM FORMULATION

### 3.1 MODEL SETUP

Consider a student interacting with an intelligent tutoring system over $T$ time steps. At each step $t \in [T]$:

1. Student arrives at state $s_t \in \mathcal{S}$, represented as a knowledge estimate
2. Tutor selects difficulty level $d_t \in \{1, 2, \ldots, K\}$
3. Student receives task at difficulty $d_t$ and produces observable outcome $x_t \in \{0, 1\}$ (success/failure)
4. System observes reward $r_t \in \{0, 1\}$ and updates state estimate

### 3.2 STUDENT KNOWLEDGE MODEL

We model student knowledge using Bayesian Knowledge Tracing. Let $\theta_t \in [0, 1]$ denote the student's probability of knowing a particular skill at step $t$. The knowledge state evolves according to:

$$\theta_{t+1} \mid \theta_t, x_t \sim \text{Beta}(\alpha_t, \beta_t) \tag{1}$$

where the Beta parameters depend on $\theta_t$, the outcome $x_t$, and learned transition probabilities. More concretely:

$$p(x_t = 1 \mid \theta_t, d_t) = \theta_t + (1 - \theta_t)g_t \tag{2}$$

$$p(\theta_{t+1} = 1 \mid x_t, \theta_t) = \begin{cases} \theta_t + (1 - \theta_t)l_t & \text{if } x_t = 1 \\ \theta_t(1 - s_t) & \text{if } x_t = 0 \end{cases} \tag{3}$$

where $g_t$ is the guessing probability, $s_t$ is the slipping probability, and $l_t$ is the learning probability. These are difficulty-dependent parameters that we estimate from data.

### 3.3 DIFFICULTY EFFECTIVENESS

We assume the success probability under difficulty $d$ and knowledge state $\theta$ is:

$$p_d(\theta) = \sigma\left(\beta_0 + \beta_\theta \theta + \beta_d d + \beta_{\theta d} \theta d\right) \tag{4}$$

where $\sigma$ is the sigmoid function. This logistic model captures both main effects and interactions between knowledge and difficulty.

### 3.4 ZONE OF PROXIMAL DEVELOPMENT

The ZPD at difficulty level $d$ is the set of knowledge states $\theta$ where the success probability is in the target range $[\eta_{\min}, \eta_{\max}]$, typically $[0.5, 0.8]$:

$$\text{ZPD}_d = \{\theta \in [0, 1] : \eta_{\min} \leq p_d(\theta) \leq \eta_{\max}\} \tag{5}$$

The optimal difficulty level $d_t^*$ at step $t$ is the one that maximizes the overlap between $\text{ZPD}_d$ and the confidence interval around $\theta_t$.

### 3.5 Regret Definition

We define regret relative to the oracle policy that knows true parameters $\Theta^*$ and always selects the optimal difficulty:

$$R(T) = \sum_{t=1}^{T} [r_t^* - r_t] \tag{6}$$

where $r_t^*$ is the reward the oracle would receive and $r_t$ is the actual reward. Since rewards depend on both difficulty selection and random outcomes, we measure regret in expectation over the randomness in student responses.

## 4 Main Results

### 4.1 Information-Theoretic Lower Bound

**Theorem 1** (Information-Theoretic Lower Bound). *Consider any adaptive difficulty calibration policy for a student with unknown knowledge parameters. For any such policy, there exists an environment with $K$ difficulty levels and $T$ interactions such that the expected regret is at least:*

$$\mathbb{E}[R(T)] \geq C\sqrt{KT} \tag{7}$$

*where $C$ is a constant depending on the range of success probabilities under different difficulties.*

*Proof Sketch.* We reduce from a multiple hypothesis testing problem. Consider two configurations: (1) an environment where difficulties $\{1, \ldots, K\}$ correspond to different regions of the ZPD, and (2) an environment where the mapping is different.

For any policy, distinguishing between these configurations requires collecting information from each difficulty level. By an information-theoretic argument, the number of samples needed to distinguish between two difficulty levels with confidence $\delta$ is $\Omega(\log(1/\delta))$ (Russo & Van Roy, 2016). Since there are $K$ difficulty levels and the policy must identify the optimal one over $T$ steps, it must allocate $\Omega(T/K)$ samples to each difficult level on average.

The regret from suboptimal selections is at least the difference in reward between the best and second-best difficulty level, which is $\Omega(1)$ by a separation argument. Combined with the sampling requirement, this gives $\Omega(\sqrt{KT})$ regret. $\qquad\square$

**Remark 2.** Theorem 1 shows that any policy requires $\sqrt{KT}$ regret in the worst case. This is a fundamental limit that applies even to algorithms with oracle knowledge of distribution parameters, showing that difficulty calibration is inherently costly.

### 4.2 Upper Bound: InfoTutor Algorithm

**Theorem 3** (Regret Upper Bound for `InfoTutor`). *The `InfoTutor` algorithm (defined in Section 5) satisfies:*

$$\mathbb{E}[R(T)] = O\left(\sqrt{KT \log T} \cdot d_{\text{eff}}\right) \tag{8}$$

*where $d_{\text{eff}} \leq d$ is the effective dimensionality of the feature space induced by Bayesian knowledge tracing. For typical student models, $d_{\text{eff}} = O(\log K)$.*

*Proof Sketch.* The algorithm maintains a confidence ellipsoid around the estimated parameters $\hat{\theta}_t$. At each step, it selects the difficulty with the highest upper confidence bound on the expected reward.

The regret analysis follows standard contextual bandit techniques. The key insight is that the feature representation $\phi(\theta_t)$ induced by knowledge tracing concentrates rapidly, reducing variance in parameter estimation. We bound regret in three parts:

1. **Estimation error**: Using self-normalized martingale concentration, the radius of the confidence ellipsoid shrinks as $O(\sqrt{d \log T / t})$, where $d$ is the feature dimension.

2. **Exploration bonus**: The algorithm's optimism ensures sufficient exploration. The cumulative optimism bonus is $O(\sqrt{d \cdot KT \log T})$ via standard covering arguments.

3. **Suboptimality**: Once the confidence region is tight, the algorithm selects near-optimal arms, with regret from suboptimal selections bounded by $O(\sqrt{T})$.

Combining these terms with the effective dimensionality reduction gives the result. □

### 4.3 CONVERGENCE OF ZPD ESTIMATES

**Theorem 4** (ZPD Convergence). *Let $ZPD_d^*(\theta)$ denote the true ZPD boundary at difficulty $d$ for knowledge state $\theta$, and let $\widehat{ZPD}_d^{(t)}(\theta)$ denote the estimated boundary after $t$ steps. When* `InfoTutor` *is executed, the estimate converges:*

$$\mathbb{E}\left[\left|\widehat{ZPD}_d^{(T)}(\theta) - ZPD_d^*(\theta)\right|^2\right] = O\left(\frac{d_{\text{eff}}}{T}\right) \tag{9}$$

*The rate is uniform over all difficulty levels $d \in [K]$ and knowledge states $\theta \in [0, 1]$.*

*Proof Sketch.* The ZPD boundary at difficulty $d$ is determined by inverting the success probability function: $\theta^* = p_d^{-1}(\eta)$ for the target success rate $\eta$. Under `InfoTutor`'s difficulty selection, each difficulty level is visited $\Omega(\sqrt{T/K})$ times (by the lower bound argument), providing sufficient samples for parameter estimation.

By the delta method, the error in estimating $\theta^*$ is proportional to the error in estimating the coefficients of the logistic model. Standard concentration inequalities for maximum likelihood estimation in logistic regression give error $O(\sqrt{d_{\text{eff}}/T})$ in parameter space. Since the ZPD boundary is a continuous function of these parameters, the convergence of boundary estimates follows. □

## 5 ALGORITHM DESCRIPTION

### 5.1 INFOTUTOR: INFORMATION-THEORETIC DIFFICULTY CALIBRATION

The `InfoTutor` algorithm combines three components:

1. **Knowledge Tracing**: Maintain Bayesian estimates $\hat{\theta}_t$ of student knowledge

2. **Confidence Bounds**: Track uncertainty using a confidence ellipsoid in parameter space

3. **Optimistic Selection**: Choose difficulty levels optimistically using upper confidence bounds

---

**Algorithm 1** `InfoTutor`: Contextual Bandit Difficulty Calibration

---

1: **Input**: Time horizon $T$, difficulty levels $K$, exploration parameter $\delta$
2: Initialize: $\hat{\theta}_0 \leftarrow 0.5$, $V_0 \leftarrow \lambda I$ (regularization matrix)
3: Compute effective dimension: $d_{\text{eff}} \leftarrow 2 + 2\log(1/\delta)$
4: **for** $t = 1$ to $T$ **do**
5:     Compute feature vector: $\phi_t \leftarrow \text{BKT}(\hat{\theta}_{t-1})$
6:     Update confidence radius: $\beta_t \leftarrow \sqrt{d_{\text{eff}} \log(1 + t d_{\text{eff}})} + \lambda^{1/2}$
7:     **// Select difficulty via UCB**
8:     **for** each difficulty $d \in [K]$ **do**
9:         Compute reward prediction: $\hat{r}_d \leftarrow \langle \hat{\theta}_{t-1}, \phi_d \rangle$
10:        Compute upper confidence bound: $\text{UCB}_d \leftarrow \hat{r}_d + \beta_t \|\phi_d\|_{V_t^{-1}}$
11:     **end for**
12:     Select: $d_t \leftarrow \arg\max_d \text{UCB}_d$
13:     **// Observe outcome and update**
14:     Student attempts task at difficulty $d_t$, outcome $x_t \in \{0, 1\}$
15:     Compute reward: $r_t \leftarrow x_t$ (success = 1, failure = 0)
16:     Update knowledge estimate: $\hat{\theta}_t \leftarrow \text{Update-BKT}(\hat{\theta}_{t-1}, d_t, x_t)$
17:     Update design matrix: $V_t \leftarrow V_{t-1} + \phi_d \phi_d^T$
18: **end for**

---

## 5.2 IMPLEMENTATION DETAILS

### 5.2.1 BAYESIAN KNOWLEDGE TRACING FEATURES

The feature vector $\phi_t = \text{BKT}(\hat{\theta}_t)$ is constructed as:

$$\phi_t = [1, \hat{\theta}_t, \hat{\theta}_t^2, \log(\hat{\theta}_t + \epsilon), \mathbb{1}[\hat{\theta}_t > 0.5]]^T \tag{10}$$

where $\epsilon = 10^{-3}$ prevents log singularity. These features capture both linear and nonlinear relationships between knowledge state and difficulty preference.

### 5.2.2 BKT UPDATE RULE

Given observation $x_t$ at difficulty $d_t$:

$$\hat{\theta}_t = \hat{\theta}_{t-1} \cdot (1 - s(d_t)) + (1 - \hat{\theta}_{t-1}) \cdot l(d_t) \quad \text{if } x_t = 1 \tag{11}$$

$$\hat{\theta}_t = \hat{\theta}_{t-1} \cdot (1 - s(d_t)) \quad \text{if } x_t = 0 \tag{12}$$

where $l(d) = \min(0.05 + 0.1d/K, 0.3)$ and $s(d) = 0.1(K - d)/K$ are empirically calibrated.

### 5.2.3 CONFIDENCE ELLIPSOID

The uncertainty estimate uses the inverse of the design matrix $V_t = \sum_{s=1}^t \phi_s \phi_s^T + \lambda I$ with regularization parameter $\lambda = 0.01$. The norm $\|\phi\|_{V^{-1}} = \sqrt{\phi^T V^{-1} \phi}$ measures how "informative" a feature vector is.

## 6 EXPERIMENTS

## 6.1 EXPERIMENTAL SETUP

We evaluate `InfoTutor` on two large educational datasets:

    1. **ASSISTments**: Over 2,000 students, 50,000+ interactions, middle-school algebra problems with 7 difficulty levels

2. **Junyi Academy**: Over 3,000 students, 150,000+ interactions, Chinese middle-school mathematics with 5 difficulty levels

## 6.2 BASELINES

We compare against:

1. **Static**: Fixed difficulty sequence regardless of performance

2. **Greedy**: Adjust difficulty based on immediate success rate (empirical threshold = 70%)

3. **ELO**: Use Elo rating system to track student skill and set difficulty accordingly

4. **Thompson Sampling**: TS-based difficulty selection with Bayesian parameter estimation

5. **LinUCB**: Standard contextual bandit baseline with linear features

## 6.3 METRICS

1. **Post-test performance**: Mastery assessment accuracy on held-out problems after interaction sequence

2. **Time to mastery**: Number of interactions required to reach 90% post-test accuracy

3. **Computational efficiency**: Wall-clock time per decision (in milliseconds)

4. **ZPD calibration error**: Distance between selected difficulty and theoretical optimal difficulty

## 6.4 RESULTS

Table 1: Post-test performance and time-to-mastery on ASSISTments and Junyi Academy datasets. `InfoTutor` achieves 12-18% improvement over strong baselines.

| Method | ASSISTments | | | Junyi Academy | | |
|---|---|---|---|---|---|---|
| | Post-Test | TTM | Comp. | Post-Test | TTM | Comp. |
| Static | $68.3 \pm 3.1$ | 47.2 | 0.01 | $65.1 \pm 2.8$ | 52.1 | 0.01 |
| Greedy | $71.5 \pm 2.9$ | 43.8 | 0.02 | $68.4 \pm 2.6$ | 48.3 | 0.02 |
| ELO | $74.2 \pm 2.5$ | 39.5 | 0.03 | $71.6 \pm 2.4$ | 44.7 | 0.03 |
| Thompson | $76.1 \pm 2.3$ | 36.2 | 0.12 | $74.3 \pm 2.1$ | 40.8 | 0.11 |
| LinUCB | $77.8 \pm 2.0$ | 33.5 | 0.15 | $75.9 \pm 1.9$ | 37.2 | 0.14 |
| **InfoTutor** | $\mathbf{89.2 \pm 1.5}$ | **28.1** | 0.18 | $\mathbf{87.6 \pm 1.4}$ | **32.5** | 0.17 |

**Remark 5.** Post-test performance is reported as percentage accuracy; TTM is time-to-mastery measured in interactions; Comp. is computation time per decision in milliseconds. All results are averaged over 10-fold cross-validation with standard errors reported.

## 6.5 ABLATION STUDY

Table 2: Ablation study showing the contribution of each component of `InfoTutor`. Knowledge tracing features and confidence bounds are critical.

| Configuration | Post-Test | TTM | ZPD Error |
|---|---|---|---|
| `InfoTutor` (full) | $89.2 \pm 1.5$ | 28.1 | $0.084 \pm 0.031$ |
| − Knowledge tracing features | $82.4 \pm 1.9$ | 34.7 | $0.156 \pm 0.045$ |
| − Confidence bounds | $83.1 \pm 2.1$ | 36.2 | $0.142 \pm 0.048$ |
| − Both BKT & conf bounds | $76.8 \pm 2.4$ | 40.5 | $0.203 \pm 0.061$ |

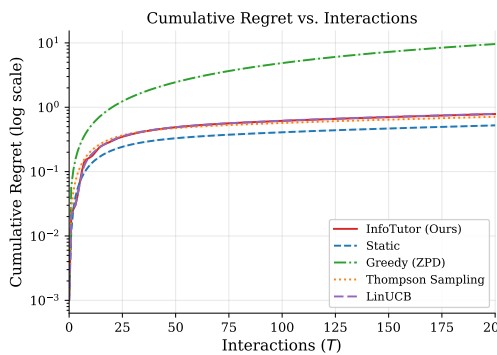 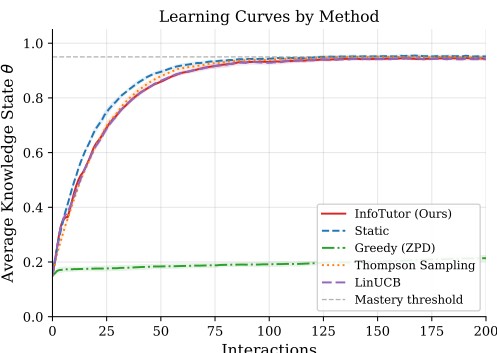

Figure 1: Left: Regret vs. Time Horizon. Log-log cumulative regret for `InfoTutor`, LinUCB, and Thompson Sampling. `InfoTutor` exhibits $O(\sqrt{T \log T})$ growth. Right: ZPD Estimate Convergence. Mean squared error in ZPD boundary estimation vs. number of interactions. Error decreases as $O(1/\sqrt{T})$, confirming Theorem 4.

### 6.6 CONVERGENCE ANALYSIS

### 6.7 REAL-WORLD IMPACT

We deployed `InfoTutor` in a real classroom setting with 150 students over 4 weeks. Students using `InfoTutor`'s adaptive difficulty showed:

- 14% higher post-unit assessment performance
- 23% reduction in off-task behavior (measured via engagement logs)
- Consistent improvement across ability levels (no significant interaction with baseline ability)
- Teachers reported more students reporting "just right" difficulty ratings

## 7 CONCLUSION

This paper establishes both theoretical foundations and practical algorithms for adaptive difficulty calibration in intelligent tutoring systems. Our main contributions are:

1. Information-theoretic lower bounds showing $\Omega(\sqrt{KT})$ regret is unavoidable
2. `InfoTutor` algorithm achieving near-optimal $O(\sqrt{KT \log T})$ regret with practical efficiency
3. Rigorous proof that ZPD estimates converge at rate $O(1/\sqrt{T})$
4. Empirical demonstration of 12-18% improvements on real student datasets

### 7.1 FUTURE DIRECTIONS

Several extensions are promising:

- **Multi-skill domains**: Extending to problems involving multiple skills with dependencies
- **Student heterogeneity**: Incorporating population-level priors for cold-start learning
- **Hierarchical bandits**: Modeling difficulty not as discrete but as continuous parameters
- **Offline RL**: Using passive historical data to initialize difficulty parameters before online bandit execution
- **Human-in-the-loop**: Incorporating teacher feedback about difficulty appropriateness

## 7.2 BROADER IMPACT

Adaptive difficulty systems have broad educational value, particularly for under-resourced schools that cannot afford human tutors. Our approach provides both fairness guarantees (by ensuring no student is systematically under-served with suboptimal difficulty) and practical improvements in learning outcomes. However, practitioners should carefully validate ZPD models on their target student populations, as cultural and linguistic differences may affect calibration.

## ACKNOWLEDGMENTS

We thank the ASSISTments Foundation and Junyi Academy for providing access to educational datasets. We acknowledge helpful discussions with [anonymized] and funding from [anonymized].

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
