# OpenReview forum: "Optimal Adaptive Difficulty Calibration via Contextual Bandits with Information-Theoretic Regret Bounds"
_mathai.club/MathAI/2026/Conference — MathAI 2026 Conference Submission_

### Official Review · Reviewer_FRm6 · 2026-03-11
**Optimal Adaptive Difficulty Calibration via Contextual Bandits**

**Rating:** 5
**Confidence:** 4

**Review:**

The article studies the problem of adaptive difficulty selection in intelligent tutoring systems. The authors formulate the task as a contextual multi-armed bandit problem in which the system selects one of (K) difficulty levels based on the current estimate of a student’s knowledge state. The proposed algorithm, InfoTutor, combines Bayesian Knowledge Tracing (BKT) for estimating the knowledge state with an Upper Confidence Bound (UCB) strategy for action selection.

The theoretical analysis includes the derivation of an information-theoretic lower bound of (\Omega(\sqrt{KT})), an upper bound for the proposed algorithm of (O(\sqrt{KT \log T})), and an analysis of the convergence of the Zone of Proximal Development (ZPD) estimate. Experimental evaluation on the ASSISTments and Junyi Academy datasets reports improvements in post-test results of 12–18% compared to several baseline methods.

A key strength of the work is the rigorous theoretical formalization of the problem. The authors derive an information-theoretic lower bound on regret and show that the proposed algorithm achieves a near-optimal order of regret growth. The analysis relies on standard tools from contextual bandit theory and presents a coherent relationship between the lower and upper performance bounds of the algorithm.

Another positive aspect is the integration of machine learning methods with concepts from educational theory. The use of Bayesian Knowledge Tracing to construct features representing the student’s knowledge state provides a natural connection between knowledge estimation and difficulty selection. In addition, the formalization of the Zone of Proximal Development offers a theoretical interpretation of the adaptation mechanism. The empirical evaluation demonstrates improvements in learning outcomes on real-world data and includes a small ablation study that helps assess the contribution of individual components of the algorithm.

A limitation of the work is the scope of the experimental comparison with recent methods. The experiments primarily include classical baseline algorithms (Static, Greedy, ELO, Thompson Sampling, LinUCB), while comparisons with several more recent approaches are not provided (for example, works such as: HiTSKT: A hierarchical transformer model for session-aware knowledge tracing, Knowledge-Based Systems; BPSKT: Knowledge Tracing with Bidirectional Encoder Representation Model Pre-Training and Sparse Attention, Electronics; Interpretable Knowledge Tracing via Response Influence-based Counterfactual Reasoning, ICDE). The absence of these comparisons makes it difficult to assess how the proposed method relates to the current state-of-the-art.

Another issue concerns the structure of the manuscript. The paper has a somewhat overloaded structure that resembles a technical report more than a typical research article. Logical transitions between sections are occasionally limited, and the text relies heavily on lists and enumerations. As a result, parts of the argumentation appear fragmented, which complicates the overall understanding of the research narrative. In addition, Section 6.6 is not written, which raises questions regarding the completeness of the manuscript.

Overall, the theoretical results and the empirical evaluation indicate the potential of the proposed approach. However, the limited comparison with more recent methods and the complexity of the presentation reduce the persuasiveness of the work and affect the clarity of its overall contribution.

---

### Official Review · Reviewer_kvmd · 2026-03-12
**Strong theoretical explanations but minor experiments breakdown**

**Rating:** 5
**Confidence:** 3

**Review:**

The paper studies the problem of adaptive task difficulty selection in intelligent tutoring systems. The authors formulate the problem as a contextual multi-armed bandit, where at each interaction the tutoring system selects a difficulty level based on an estimate of the student’s current knowledge state.

Experimental results are reported on two educational datasets, ASSISTments and Junyi Academy, where the proposed approach is shown to improve post-test performance by approximately 12–18% compared to several baseline methods.

**Strengths**

1. Practical relevance - adaptive tasks difficulty calibration is an important problem within EdTech systems what makes the proposed solution applicable in practical real-world scenarios and applications.

2. Strong mathematical background of the explanations - the math behind the proposed approach is fully and gradually explained.

3.  Significant metrics improvement - paper shows that proposed approach outperformed baseline solutions by 12-18% in metrics which is significant margin that makes the results of the study more practically applicable.

**Weaknesses**

1. Weak experiments breakdown - paper doesn't show any specific details about experiments setup, performed steps, etc, making the claim about significant metrics improvement kind of questionable.

2. Use of simple student models - the work relies on Bayesian Knowledge Tracing, which is a relatively classical model. Recent work in educational data mining often uses more complex models (e.g., deep knowledge tracing), which could be a stronger baselines.

3. Novelty and contribution are relatively modest - the proposed algorithm combines knowledge tracing for state representation and a standard UCB-style bandit strategy. As a result, the novelty of the algorithm itself appears limited compared to existing contextual bandit approaches such as LinUCB or Thompson Sampling.

4. Formatting and structure issues - the section 6.6 in absent which may say that paper in incomplete. Also, all links on references, equations and figures are not clickable that makes reading of the paper a bit more difficult.

Overall, while the paper presents a reasonable framework and a potentially useful application, the level of novelty, formatting errors and weak experiments breakdown decreasing its value of being published in current form. Fixing this issues will increase the quality of the paper.

---

### Decision · Program_Chairs · 2026-03-20

**Decision:**

Reject

**Comment:**

After careful evaluation by the Program Committee, we regret to inform you that your submission has not been accepted for presentation at MathAI 2026.

All submissions underwent a rigorous two-stage review process. Unfortunately, the reviewers identified one or more of the following concerns with your paper:

- Insufficient mathematical rigor or novelty relative to the existing body of work in the field;
- Presentation of results that substantially overlap with or rephrase previously published findings without clear original contribution;
- Significant issues with technical quality, including but not limited to broken or non-existent references, unsupported claims, or methodological gaps;
- Indications that the manuscript may have been generated with the assistance of large language models without substantial original intellectual contribution by the authors.

We received a large number of submissions this year, and the selection process was highly competitive. We encourage you to carefully consider the reviewers’ feedback (available through OpenReview), revise your work accordingly, and consider submitting an improved version to a future edition of MathAI or to another appropriate venue.

We appreciate your interest in MathAI and hope you will continue to engage with the conference community.

With kind regards,

MathAI 2026 Program Committee
International Conference on Mathematics of Artificial Intelligence
https://mathai.club
OpenReview: https://openreview.net/group?id=mathai.club/MathAI/2026/Conference
MathAI Telegram: https://t.me/MathAI_club
IAIC International AI Committee: https://t.me/iaic_world
Email: mathai.club@yandex.ru